# Research Progress of Fibroblasts in Human Diseases

**DOI:** 10.3390/biom14111478

**Published:** 2024-11-20

**Authors:** Xiaodong Li, Nana Li, Yujie Wang, Qixiang Han, Boshi Sun

**Affiliations:** Department of Colorectal Surgery, The Second Affiliated Hospital of Harbin Medical University, Harbin 150086, China; xiaodong.li@hrbmu.edu.cn (X.L.); nana.li@hrbmu.edu.cn (N.L.); wangyj@hrbmu.edu.cn (Y.W.); qixiang.han@hrbmu.edu.cn (Q.H.)

**Keywords:** fibroblasts, wound healing, fibrosis, cancer, cancer associated fibroblasts, therapy

## Abstract

Fibroblasts, which originate from embryonic mesenchymal cells, are the predominant cell type seen in loose connective tissue. As the main components of the internal environment that cells depend on for survival, fibroblasts play an essential role in tissue development, wound healing, and the maintenance of tissue homeostasis. Furthermore, fibroblasts are also involved in several pathological processes, such as fibrosis, cancers, and some inflammatory diseases. In this review, we analyze the latest research progress on fibroblasts, summarize the biological characteristics and physiological functions of fibroblasts, and delve into the role of fibroblasts in disease pathogenesis and explore treatment approaches for fibroblast-related diseases.

## 1. Introduction

Fibroblasts (Fbs) are one of the most prevalent cell types in the human body and the most important cell types in loose connective tissue. They are widely distributed in skin, muscle, bone, liver, lung, heart, kidney, blood vessels, and other tissues and organs. Fbs have a strong metabolism and a strong ability of proliferation, and they have the ability to differentiate into different cell types when exposed to specific induction conditions, thus showing their strong potential of multi-directional differentiation. Mesenchymal stem cells are a kind of pluripotent stem cells with strong self-renewal and differentiation abilities. Fbs and mesenchymal stem cells have many similarities, and it is difficult to distinguish these two cell types in terms of morphology, surface markers, proliferation, differentiation, and immunomodulatory abilities [1]. But there are some differences between Fbs and mesenchymal stem cells in gene expression profiles and epigenetic patterns, which can be used to distinguish and identify Fbs and mesenchymal stem cells [2]. Due to the similarities between Fbs and mesenchymal stem cells in many aspects, some studies have speculated that Fbs may be differentiated or senescent mesenchymal stem cells [3]. There are also some cells called fibroblast-like cells. They are similar to Fbs in morphology and function, but there are significant differences in origin and phenotype between them and Fbs. The source of fibroblast-like cells is more extensive than that of Fbs, and at the same time, they also have greater plasticity and adaptability [4,5,6].

Fbs have a variety of important biological functions. First of all, Fbs can synthesize and release extracellular matrix (ECM) components, including collagen and elastic fibers, thus providing tissues and organs with structure and elasticity. In addition, Fbs also show immunomodulatory function, which can participate in the regulation of immune responses and immune tolerance by regulating the activity and differentiation of T cells [7]. Fbs can also secrete many kinds of cytokines and growth factors, and participate in wound healing, immune responses, angiogenesis, and other physiological processes, Fbs play an essential role in biological growth and development, along with tissue repair [8]. Fbs play a crucial part in the formation of tissues and organs through active migration and proliferation in the developmental stage. In the mature stage, Fbs enter into a static or dormant state; however, they still maintain the function of synthesizing extracellular matrix components and maintaining tissue homeostasis. When the body is injured, they will resuscitate and migrate directionally to the injured site, thus promoting healing, fibrosis, and tissue repair. In addition to playing a role in normal tissue function, Fbs also participate in several pathological processes, as well as play a significant part in the occurrence and progression of fibroblast-related diseases. These diseases mainly include fibrosis, cancers, and some autoimmune diseases. In this review, we summarize the biological characteristics and physiological functions of Fbs, as well as delve into the role of fibroblasts in disease pathogenesis and explore treatment approaches for fibroblast-related diseases.

## 2. Biological Characteristics of Fbs

### 2.1. Morphological and Structural Characteristics and Cellular Markers

Fbs were first discovered by Rudolf Virchow in 1858, when he referred to them as “spindle-shaped cells of the connective tissue”. Ernst Ziegler later coined the term “fibroblast” to refer to cells responsible for depositing new connective tissue during the healing process [8]. Currently, there is no accepted standard definition of Fbs, and the word “fibroblasts” is only a definition of it when regarding morphology. There are a large number of and widely distributed Fbs in the human body. When the function of Fbs is active, the cells are larger with more protuberances, exhibiting antlers or stars, large and regular oval nucleoli, obvious nucleoli, and rich cytoplasm, which demonstrate weak basophilia. Under electron microscopy, abundant rough endoplasmic reticulum, Golgi complexes, and free ribosomes can be observed in the cytoplasm of Fbs, thus indicating that Fbs have the ability to synthesize and secrete proteins (Figure 1). When the function of a fibroblast is in a static state, it is referred to as a fibrocyte. At this time, the cell is small in size and exhibits a long fusiform; moreover, the nucleus is small and slender, the nucleolus is not obvious, and there is less cytoplasm, whereby it shows eosinophilic characteristics. Under electron microscopy, the rough endoplasmic reticulum and Golgi complex in fibrocytes have been shown to be underdeveloped.

Cell markers are often used to distinguish and isolate different types of cells; however, at present, no specific markers for Fbs have been found. Currently, some common markers of Fbs have been identified, such as fibroblast-specific protein 1 (FSP1), platelet-derived growth factor receptor (PDGFR), vimentin, CD44, CD73, CD90, CD105, and other markers [3,9,10,11,12]. However, these markers are not unique to Fbs, and not all Fbs can express all of these markers. For example, PDGFRα is also expressed in cells of the central nervous system, and PDGFRα is not expressed in some types of Fbs [13,14]. In addition, Fbs have a certain heterogeneity, and their functions and cellular markers will be different according to the surrounding tissue environment [8].

### 2.2. Subtype Heterogeneity and Tissue Heterogeneity

With the emergence of single-cell RNA sequencing (scRNA-seq) technology, different subtypes of Fbs and the various functions of these subtypes are gradually becoming known to the public. Traditional cell sequencing technologies usually perform whole-genome sequencing on a group of cells, and such results can mask the subtle differences between individual cells. The scRNA-seq technology can perform transcriptome analysis on a single cell, identify Fb subpopulations with different functions in various tissue types, and reveal the diversity and heterogeneity between cells. Fbs have a certain degree of heterogeneity. According to the different surrounding tissue environments they are in, their functions and cell markers will also be different [8]. By combining scRNA-seq with other spatial transcriptomics methods, it was found that different subpopulations of Fbs in the same organ have different roles and functions [15,16,17,18,19]. For example, through the analysis of lung Fbs and cardiac Fbs in mice by using scRNA-seq technology, it was found that lung Fbs can be divided into three subpopulations according to different locations and specific markers, and cardiac Fbs can be divided into five subpopulations according to different functions [15,16] (Table 1). However, there may be other classifications of subpopulations in human lung Fbs and cardiac Fbs [15,18,20]. Kuppe et al. [18] performed a clustering analysis of human cardiac Fbs by means of scRNA-seq technology. It was found that cardiac Fbs can be divided into four sub-clusters (Fib1-4). Moreover, each sub-cluster has distinct features. For instance, Fib1 can be labeled by scavenger receptor class A member 5 (SCARA5), and Fib2 can be marked by periostin (POSTN), collagen type I alpha 1 (COL1A1), and fibronectin 1 (FN1).

In different tissues, there are similarities and tissue-specific characteristics between fibroblast subpopulations [21,22,23]. Gao et al. [21] performed scRNA-seq technology analysis on fibroblasts from 517 human samples and identified 20 transcriptionally distinct fibroblast clusters (FCs). In the fibroblast atlas, nine “universal” FCs and six “tissue-specific” FCs were found (Table 2).

### 2.3. Differentiation and Proliferation Capacity

Fbs originate from mesenchymal cells during embryological development. Most Fbs in the human body differentiate from mesenchymal cells in paraxial mesoderm and lateral mesoderm; however, Fbs in craniofacial tissue differentiate from neural crest mesenchymalcells [24]. Most cardiac Fbs are derived from epicardial epithelial cells through epithelial–mesenchymal transformation (EMT) and endocardial epithelial cells through endothelial–mesenchymal transformation (EndMT) [25]. In the event of inflammatory injury and trauma, epithelial cells can also be transformed into Fbs and other related cell types through the EMT process, thus improving the ability of migration for tissue reconstruction [26]. In addition, mesenchymal stem cells from the placenta, umbilical cord blood, bone marrow, and fat can produce Fbs under certain induction conditions [27].

Fbs need to maintain their normal function via proliferation and self-renewal. The platelet-derived growth factor (PDGF) signal pathway is the main signal pathway of Fb proliferation and self-renewal [28]. PDGF ligands can be produced by Fbs or other types of cells, and four different PDGF polypeptide chains are connected by disulfide bonds to form four homodimers and a heterodimer, in which PDGF-AA, PDGF-BB, and PDGF-AB are processed and released in the form of dimers, whereas PDGF-CC and PDGF-DD are released in an inactive form [29,30]. PDGF ligands bind to the corresponding tyrosine kinase receptors and activate several downstream signal cascade pathways, such as RAS/MAP and PI3K/AKT pathways, thus transmitting the signals to downstream effectors to exert their effects.

Siebuhr et al. [31] found that the proliferation of Fbs was enhanced after being treated with PDGF. A study by Wang et al. [32] showed that PDGF-BB can promote Fb proliferation, migration, and collagen synthesis through the PDGFR-β/PI3K/AKT signal pathway. When the PDGF-A ligand was lost, the proliferation and self-renewal ability of lung Fbs was significantly decreased [33]. Bayer et al. [34] found that PRGF is capable of mediating the induction of ECM-related factors in fibroblasts. Therefore, the PDGF signaling pathway assumes a leading role in proliferation, self-renewal and long-term dynamic balance of Fbs.

Fibroblast growth factor (FGF) plays a significant role in the proliferation, migration, and differentiation of Fbs. In most instances, the binding of FGF to fibroblast growth factor receptor (FGFR) requires heparan sulfate (HS) glycosaminoglycan as a co-receptor to form a ternary complex, thereby ensuring the stable binding of FGF and FGFR [35]. Once FGF binds to FGFR, it activates downstream signaling pathways to exert its function. The main signaling pathways include RAS/mitogen-activated protein kinase (MAPK), PI3K–AKT, phospholipase C gamma (PLCγ), and signal transducer and activator of transcription (STAT), etc. [36]. FGF1 can promote the aggregation and proliferation of Fbs at the damaged site and promote the differentiation of fibroblasts into myofibroblasts by inducing the secretion of transforming growth factor-β (TGF-β) by damaged skin tissues, thereby promoting the deposition of ECM and wound contraction activities, and finally forming granulation tissue [37]. FGF2 has been confirmed to promote Fb proliferation and can promote collagen synthesis, which is crucial for wound healing and tissue regeneration [38].

TGF-β signaling can induce the activation of Fbs into myofibroblasts, mainly through the classical TGF-β/SMAD signaling pathway [39]. Non-classical TGF-β signal transduction can also participate in the process of myofibroblast transdifferentiation. After skin fibroblasts are stimulated by TGF-β signals, the phosphorylation level of MAP kinase ERK1/2 will increase, and then the transcription factor FRA2 will be upregulated, promoting the differentiation of Fbs into myofibroblasts [40]. TGF-β signaling plays an important role in wound healing and fibrosis.

## 3. Physiological Function of Fbs

### 3.1. Organizational Structure

The formation and maintenance of the structures of most organs and tissues are largely dependent on Fbs [8]. Fbs have the ability to synthesize and secrete ECM proteins such as collagen, elastin, and fibronectin, in which type I collagen forms collagen fibers, type III collagen forms reticular fibers, and elastin forms elastic fibers. Collagen increases the strength and stability of tissues, and elastin increases the elasticity and flexibility of tissues, as well as providing the extracellular matrix with some tolerance to mechanical stress [41]. But elastin production in fibroblasts is only during perinatal phase and not seen in adult type of cells, where it is very limited [42]. In addition to synthesizing proteins, Fbs can also synthesize and secrete hyaluronic acid (HA), proteoglycans, and other components to maintain the gel state and hydration of the extracellular matrix, thus facilitating cell migration and extracellular matrix remodeling [43]. Together, these proteins and molecules form a complex extracellular matrix, which plays an external supporting role around the cells, thus maintaining the stability of tissue structure and function [44].

Fbs control matrix protein degradation by secreting ECM-modifying enzymes, including matrix metalloproteinases (MMPs) and tissue inhibitors of matrix metalloproteinases (TIMPs), thus remodeling the microstructure of the ECM [45]. When Fbs reshape tissues, the components in the ECM are combined according to a certain proportion and perform subtle tissue remodeling activities, which can produce an ECM with different components, microanatomical structures, and physiological functions in different organs; moreover, these ECM components can support their corresponding cells, such as elastic and soft skin keratinocytes, as well as elastic and malleable lung epithelial cells, among other cells [46].

### 3.2. Wound Healing

Unlike the highly proliferative state in the embryonic stage, Fbs mostly remain static after birth. Although the resting state of Fbs can last for a long period of time, Fbs are crucial to the process of wound healing [24]. The skin functions as the primary barrier shielding the body from the external environment. Skin Fbs are the main cell groups in the dermis and have the ability to repair wounds caused by daily wear, as well as burns and lacerations from the environment, in order to maintain the protective effect of the skin. Skin Fbs proliferate massively via mitosis; additionally, they synthesize and release abundant collagen fibers and matrix components in conjunction with new capillaries, contribute to the formation of granulation tissue, as well as repair defects in damaged tissue and create necessary conditions for epidermal cells coverage [47]. In visceral injury, most of the Fbs that are involved in the repair process come from the stroma and capsule, as well as connective tissue in the submucosa or subserosa.

Wound healing generally is divided into three stages, including inflammation, repair, and remodeling; Fbs are important during each of these phases. In the early stage of injury, platelets and inflammatory cells guide Fbs to migrate to the injured site, after which Fbs repair the skin skeleton by synthesizing and depositing new ECM [24]. Fbs secrete fibronectin and can transition into activated myofibroblasts in response to mechanical tension and certain cytokines presented by TGF-β, thus secreting collagen and expressing α-smooth muscle actin, followed by neovascularization and finally granulation tissue [48]. In the tissue remodeling stage of wound healing, myofibroblasts begin to contract and rearrange the deposited collagen and other ECM components to change the structure of the granulation tissue. Over time, the ECM is gradually strengthened and stiffened, thus resulting in scar tissue [49].

Different Fb subsets have differences in migration and function, and the surrounding microenvironment and cell-to-cell interactions also affect the function of Fbs. It has been found that there are different phenotypes of Fbs during the process of wound healing, and mechanical factors, cell–cell interactions, mechanical tension, and age can also affect the behavior and function of Fbs [47]. The pedigree tracing of Fbs has shown that, at the initial stage of injury, injury repair was initially mediated by subdermal reticular Fbs to form the extracellular matrix, whereas Fbs in the dermis played a role in tissue remodeling and regeneration [50]. In the later stage of wound healing, Fbs take part in tissue remodeling after repair by secreting collagenase.

The migration patterns of different Fb subtypes are also different. Interfollicular Fbs are a type of Fbs responsible for filling the wound bed. Their migration does not require the support of cell–cell communication and can be performed alone. However, fascia Fbs move in a similar manner to bees when they migrate to the wound bed. Fascia Fbs mainly increase the contacts of cell–cell communication by up-regulating the intercellular connexins N-cadherin and Cx43, in order to obtain better cell support and intercellular interaction [51,52]. This bee-like migration pattern of fascia cells allows the local ECM to be pulled into the wound and accelerate wound closure [53]. However, when Cx43 fails to function, the migration of fascial Fbs and the deposition of ECM into the wound will be inhibited, which may eventually lead to scar formation. Bee-like collective cell migration is a unique cell migration mode of fascial Fbs. During the process of wound repair and scar formation, in order to complete bee-like cell migration, the expression of N-cadherin must be upregulated, whereas there is no N-cadherin in the upper skin and oral mucosa; thus, there is no bee-like activity in wound repair, and the scar is the smallest [51].

Goss et al. [54] found that the lineage of Lrig1+ papillary or Dlk1+ reticular Fbs contribute to the production of perivascular cells during the process of angiogenesis. Fbs also have the function of promoting hair follicle and adipocyte regeneration. The formation of hair follicles requires the participation of Wnt/β-catenin signals in the epidermis, and this process is often accompanied by the extensive remodeling of the dermal ECM [55,56]. Bone morphogenetic protein (BMP) and insulin-like growth factor are involved in the expression of keratinocyte-specific β-catenin, thus leading to an increase in adipogenesis [57]. Additionally, the transforming growth factor-β (TGF-β) signal is vital for wound repair and scar formation [58].

Cell-to-cell interactions may affect the behavior of Fbs. Transcriptome analysis demonstrated that acid-sensing ion channel 3 (ASIC3)- macrophage colony-stimulating factor (M-CSF) signal promotes the polarization of M2 macrophages, thus resulting in an increase in TGF-β1 content, which eventually mediates Fbs to differentiate into myofibroblasts [59]. In addition, Fbs also respond to some mechanical pressures, such as the Yes-associated protein (YAP) signal in the Hippo pathway that can regulate the stiffness of the ECM [60,61].

Mechanical force serves as a critical regulator of Fb function and behavior, exerting significant influences on tissue homeostasis maintenance, wound healing, and pathological responses. When Fbs are exposed to mechanical stimuli, such as stretching, compression, or shear forces, they can modulate their cell stiffness by adjusting the cytoskeleton or enhancing adhesion between the Fbs and the ECM. Appropriate mechanical forces can alter Fb morphology, facilitating a better adaptation to the external physical environment. Concurrently, these forces enhance the fibroblasts’ capacity to synthesize and secrete ECM components, thereby providing structural support for tissues and maintaining tissue stability [62,63]. Fibroblasts can receive and respond to mechanical signals within the extracellular environment via diverse cell-surface receptors, such as integrins, G-protein-coupled receptors (GPCRs), and stretch-activated ion channels (SACs) [64]. This process profoundly impacts Fb proliferation, migration, and differentiation. It is of utmost significance in tissue-damage repair. Under mechanical force stimulation, Fbs can synthesize and secrete ECM components, such as collagen, which furnishes a structural scaffold for nascent tissues, augments tissue strength, and promotes wound healing [65]. Furthermore, appropriate mechanical forces can align Fbs and the ECM in a particular direction, resulting in more orderly wound healing and enabling the repaired tissue to more closely resemble normal tissue in both function and structure.

### 3.3. Tissue Regeneration

Fbs are the main participants in tissue regeneration, which is mainly due to the interaction between Fbs and stem cells and the strong transdifferentiation potential of Fbs.

Currently, many studies have confirmed that Fbs and stem cells can interact with each other, as was evidenced when Fbs and stem cells were co-cultured, and they may influence the proliferation and differentiation of stem cells. Li et al. [66] found that, in the co-culture of Fbs and human bone marrow-derived mesenchymal stem cells, the capacity of mesenchymal stem cells to differentiate into vascular smooth muscle cells was enhanced. Moreover, the transcription of labeled genes and the expression of contractile device proteins were enhanced, and the phenotypic functional activity of vascular smooth muscle cells was enhanced. This indicates that Fbs can directly induce mesenchymal stem cells derived from human bone marrow to differentiate into mature and functionally active vascular smooth muscle cell phenotypes under certain conditions. In addition, periodontal ligament Fbs can secrete Fbs to participate in MAPK-mediated signaling pathways, thus regulating the proliferation of embryonic stem cells and their osteogenic differentiation [67].

Although most Fbs originate from mesenchymal cells in the embryonic mesoderm, Fbs can differentiate not only into cells originating from mesoderm but also into cells originating from endoderm and ectoderm under a specific induction medium. Takahashi et al. [68] transferred four factors into the Fbs of mouse tail skin with adenovirus vector and obtained induced pluripotent stem cells (iPSCs) via reprogramming. In another study, after human skin Fbs were transformed into iPSCs, the Fbs acquired differentiation ability, which could induce a variety of cells to differentiate [69]. As induced by human hemagglutinin, dermal Fbs can transdifferentiate into myofibroblasts and secrete ECM components [70]. Ichim et al. [71] found that dermal Fbs inoculated in a bioreactor composed of calcium alginate beads can differentiate into chondrocytes under hypoxia induction. Zhang et al. [72] found that DNA dioxygenase Tet3 can mediate the demethylation of DNA, thus transforming Fbs into functional neurons and expressing markers of mature neurons. Lee et al. [73] reprogrammed human dermal Fbs and transdifferentiated Fbs into endothelial cells via the overexpression of ER71/ETV2. Lysy et al. [74] found that Fbs can differentiate into hepatocytes in vitro, and the differentiated hepatocytes also have the function of storing glycogen and synthesizing urea, whereas, during in vivo experiments, Fbs can express cellular markers of hepatocytes when they are transplanted into damaged liver tissue. However, only a few Fbs have multi-directional differentiation potential [75,76].

## 4. The Mechanism of Fbs in Diseases

### 4.1. Fibrosis

Fibrosis refers to the abnormal repair response of tissue when some serious trauma, repetitive injury, iatrogenic injury, or disease occur. At present, it is believed that the disease burden caused by fibrosis is considerable, and common diseases such as idiopathic pulmonary fibrosis (IPF), heart failure, liver cirrhosis, scleroderma, diabetic nephropathy, hypertensive nephropathy, chronic pancreatitis, and inflammatory bowel disease have been found to be associated with fibrosis [77] (Figure 2).

As the main cell types regulating ECM homeostasis, Fbs are essential to the fibrosis process [78]. The following sections will focus on the involvement of Fbs during the process of skin, lung, and liver fibrosis. The interaction between Fbs and macrophages can also regulate the process of fibrosis. Fbs can induce macrophages to migrate to fibrotic lesions by providing the growth factor CSF1 and chemokine CCL2, as well as activate macrophages by secreting Interleukin-6 (IL-6) and TGF-β, whereas activated macrophages can secrete molecules such as IL-6, TGF-β, PDGF, and AREG. These molecules can induce the proliferation or activation of Fbs and promote the development of fibrosis [79,80].

Mechanical force plays an important role in the occurrence and development of fibrosis. Mechanical force activates intracellular signaling pathways by regulating the mechanical transduction pathways of Fbs, leading to massive proliferation of Fbs and the generation of excessive ECM components, ultimately resulting in the excessive deposition of ECM components [81]. In fibrosis of organs such as the lung, liver, and heart, excessive mechanical stress can induce the activation of Fbs into myofibroblasts. These activated cells synthesize and secrete more ECM components, which eventually contribute to fibrosis formation [82].

The outcome of fibrosis depends on the location and severity of the damage. For the fibrosis of the skin, the skin may thicken and deform, whereas for the fibrosis of the lung, liver, heart, kidney, and other internal organs, it may lead to tissue structure destruction, organ dysfunction, or even failure; in severe cases, it can possibly lead to death. The common circulating biomarkers of fibrosis in different organs will also be different (Table 3) [83,84,85,86,87,88,89,90,91,92,93,94,95,96,97,98,99,100,101,102,103,104,105,106,107,108,109]. The following will focus on the process of Fbs participating in skin, lung, liver, heart, and kidney fibrosis.

#### 4.1.1. Skin Fibrosis

When the skin is damaged by mechanical trauma, thermal burns, infections, and surgical operations, among other causes (especially when repetitive injury and chronic inflammatory disease occur), the repair function of tissues loses its normal regulation; moreover, a large number of Fbs are activated, and excessive ECM components are produced. ECM components (such as collagen and fibrin) are excessively deposited into the damaged tissue. Eventually, fibrosis occurs in the damaged skin tissue. When skin fibrosis occurs, the damaged area of the skin may thicken and deform, forming hypertrophic scars and keloids, thus affecting skin regeneration and repair, as well as normal skin function [110]. Scleroderma, also known as systemic sclerosis, is one of the skin fibrosis diseases. It is characterized by thickened and hardened skin. Moreover, fibrosis can involve multiple internal organs. It is a rare autoimmune disease with a complex etiology and a poor prognosis [111].

There are multiple signaling pathways in the skin that mediate the formation of fibrosis. The proliferation and activation of Fbs play an important role in the process of skin fibrosis. Some studies have shown that the TGF-β/Smad pathway [112], MAPK pathway [113], and Wnt/β-catenin pathway [114] can induce Fb proliferation to produce excessive ECM components, thus leading to a significant quantity of ECM deposition and skin fibrosis.

#### 4.1.2. Pulmonary Fibrosis

The most common pulmonary fibrosis disease is IPF. IPF has a high mortality and poor prognosis. The median survival time of patients without lung transplantation is less than 4 years [115]. Under normal physiological conditions, Fbs are usually distributed in the space between the alveoli and capillaries, which is an important part of lung tissue cells. Pulmonary fibrosis is caused by the proliferation and aggregation of pulmonary Fbs, accompanied by the production and deposition of excessive ECM components, thus resulting in the destruction of alveolar tissue for gas exchange and ultimately the destruction of the lung structure. Fibroblastic foci represent an important pathological feature of pulmonary fibrosis, and the progression of pulmonary fibrosis is likely to be intimately associated with the increase in fibroblastic foci.

The occurrence and development of pulmonary fibrosis are regulated by many signal pathways, including the TGF-β/Smad signal pathway, Wnt/β-catenin signal pathway, PDGF signal pathway, phosphoinositide-3-kinase (PI3K)-protein kinase B (Akt) signal pathway, vascular endothelial growth factor (VEGF) signal pathway, and the fibroblast growth factor (FGF) signal pathway. The TGF-β/Smad signal pathway is an important signal pathway for activating myofibroblasts and depositing the ECM when tissue fibrosis occurs. The inhibition of the TGF-β/Smad signal pathway has shown promise in ameliorating fibrosis, and it may represent a potential therapeutic target for pulmonary fibrosis treatment [116,117]. The Wnt/β-catenin signal pathway is important to pulmonary inflammation and fibrosis. The Wnt/β-catenin signal pathway may increase ECM deposition by inducing EMT and promoting Fb proliferation, thus participating in pulmonary fibrosis [118]. Guan et al. [119] showed that bone morphogenetic protein 4 (BMP4) was negatively correlated with fibrosis genes and could inhibit the activation of the Smad1/5/9 signal pathway and Smad2/3 signal pathway in Fbs. Liu et al. [120] found that, in normal lung tissue, the activation of the HER2 signal pathway can increase the activity of Fbs and promote pulmonary fibrosis, whereas antagonizing the HER2 signal pathway can inhibit Fb invasion and improve pulmonary fibrosis. Tsoyi et al. [121] found that silencing of CD148 can increase the production of the ECM and enhance the resistance to apoptosis in human IPF lung Fbs, whereas the overexpression of CD148 can improve pulmonary fibrosis; therefore, the activation of CD148 can play a certain role in anti-fibrosis.

#### 4.1.3. Liver Fibrosis

Liver fibrosis is a type of repair imbalance caused by repeated and continuous damage to liver tissue induced by various factors. The aggravation of liver fibrosis will gradually develop into liver cirrhosis, which may be accompanied by serious complications, such as portal hypertension. During the pathogenesis of liver fibrosis, hepatic stellate cells play a crucial role in the pathogenesis of liver fibrosis as they are the primary origin of myofibroblasts [122]. Under normal circumstances, hepatic stellate cells are always at rest. When the liver tissue is repeatedly or continuously injured, the hepatic stellate cells in the damaged liver tissue are activated and transformed into myofibroblasts, thus producing and secreting a significant quantity of ECM components, which promotes the formation of liver fibrosis [123]. During this process, there is a reconstruction of the intrahepatic structure, leading to the gradual replacement of normal liver parenchyma with fibrous scaffolds comprising collagen fibers and other ECM proteins in the ECM, thus resulting in liver dysfunction.

The activation and proliferation of hepatic stellate cells into myofibroblasts play a pivotal role in the occurrence and progression of liver fibrosis. This process is intricately regulated by a variety of signal pathways. The TGF-β/Smad3 signal pathway is one of the most significant of them. This signal pathway may stimulate the activation and proliferation of hepatic stellate cells, encourage the production and deposition of ECM components, and lead to liver fibrosis [124]. The Wnt/β-catenin signal pathway mainly promotes liver fibrosis by cooperating with the TGF-β/Smad3 signal pathway [125,126]. Ramachandran et al. [127] analyzed the transcriptomes from healthy and cirrhotic human livers and found that signal pathways, such as the TNF Receptor Superfamily Member 12A (TNFRSF12A) pathway, Notch pathway, and PDGFR pathway, can promote fibrosis. In addition, Yes-associated protein (YAP) in the Hippo signal pathway has additionally been discovered to be connected to liver fibrosis mediated by hepatic stellate cells [128].

#### 4.1.4. Cardiac Fibrosis

Cardiac fibrosis refers to the activation of cardiac Fbs into myofibroblasts under the action of various stimulating factors, generating excessive ECM components, leading to an imbalance between the production and degradation of ECM. Transient or mild fibrosis is very important for maintaining the integrity of cardiac structure. However, continuous or severe fibrosis will lead to an excessive deposition and abnormal distribution of ECM components and cause cardiac interstitial remodeling, resulting in decreased myocardial elasticity, formation of scar tissue, and ultimately lead to myocardial contractile dysfunction, reduced cardiac output, and cause heart failure [129]. Heart failure is characterized by a high morbidity and mortality [130]. The basis of cardiac fibrosis formation is the activation of cardiac Fbs. And the activation of cardiac Fbs involves many neurohumoral pathways. The renin–angiotensin–aldosterone system (RAAS) mainly activates Fbs through the angiotensin II/angiotensin receptor 1 (AT1) axis and aldosterone, and is the main way leading to cardiac fibrosis [131]. Beta-adrenergic can activate cardiac Fbs and induce fibrosis in the heart, ultimately leading to myocardial remodeling and heart failure. Inhibiting the activation of beta-adrenergic receptors can block myocardial remodeling. Beta-blockers are also commonly used drugs for treating heart failure in clinical practice [132,133,134]. Chemokines CCL2 and CXC chemokines are important regulatory factors in the process of cardiac fibrosis [135,136]. TGF-β can induce Fbs to transform into myofibroblasts and increase the synthesis of ECM proteins [137]. Some pro-inflammatory cytokines are correlated with cardiac fibrosis. Significantly increased levels of tumor necrosis factor-α (TNF-α), interleukin (IL)-1β, and IL-6 are found in the process of cardiac fibrosis [138,139,140]. In addition, PDGF, the Wnt/β-catenin axis, and other factors also participate in the activation of Fbs [141].

#### 4.1.5. Kidney Fibrosis

Kidney fibrosis is a dynamically developing process. When various factors cause kidney damage, renal Fbs proliferate excessively and are activated into myofibroblasts, generating a large amount of ECM components. The excessive deposition of ECM components generates kidney fibrosis, leading to kidney tissue damage and kidney dysfunction. Kidney fibrosis is also the main pathological feature and ultimate outcome of chronic kidney diseases [142]. The activation of kidney myofibroblasts is the main event in the occurrence and development of kidney fibrosis. At present, single-cell sequencing analysis has found that there are mainly three sources of myofibroblasts in the human kidney: PDGFRα–PDGFRβ+RGS5+NOTCH3+ pericytes; PDGFRα+PDGFRβ+MEG3+ fibroblast; and PDGFRβ+COLEC11+CXCL12+ fibroblast [143]. TGF-β signaling plays a key regulatory role in the activation of myofibroblasts and the occurrence and progression of kidney fibrosis [144].

### 4.2. Cancer

#### 4.2.1. Cancer-Associated Fibroblasts (CAFs)

CAFs are dispersed within the cancer matrix, directly interacting with cancer cells, and contributing to the occurrence and progression of cancers through a variety of mechanisms [145]. CAFs can regulate the process of cancer proliferation and metastasis, immune escape, drug resistance, and remodeling of the ECM. CAFs play a dual role in cancer development and metastasis, but they mainly elicit a promotion effect [146,147,148].

CAFs are a group of cells with high heterogeneity; moreover, there are various sources of CAFs, and the biomarkers and functions of CAFs show a certain degree of heterogeneity. The exact origin of CAFs is unclear; however, CAFs can be transformed from various cell types, such as the direct activation of resident Fbs and stellate cells, the EMT, EndMT, pericyte transdifferentiation, adipocyte transdifferentiation, smooth muscle cell transdifferentiation, as well as the recruitment and activation of mesenchymal stem cells [149]. At present, no specific cell markers of CAFs have been found. Based on their various effects, the cell markers of CAFs can be classified into three groups, including CAF markers with cancer promotion effects, CAF markers with cancer-restraining effects, and CAF markers with bidirectional effects [150] (Table 4). Through single-cell RNA sequencing and multiple imaging techniques, it was found that there are different CAF subtypes in different types of cancer. CAFs can be divided into four categories, including iCAFs (inflammatory CAFs), apCAFs (antigen presentation CAFs), myCAFs (myofibroblast CAFs), and vCAFs (vascular CAFs) [151].

#### 4.2.2. Interaction Between CAFs and the Tumor Microenvironment

The tumor microenvironment (TME) has become an important concept in oncology research in recent years, and it plays a significant part in the occurrence, progression, and metastasis of cancers [152]. CAFs serve as the primary source of cytokines, exosomes, and growth factors within the TME. CAFs mainly communicate with cancer cells and other stromal cells through the secretion of a diverse array of cytokines, chemokines, exocrine signals, and metabolites that are associated with promoting carcinogenesis and invasion, thereby remodeling the ECM, regulating local immune function, and promoting neovascularization, thus regulating cancer proliferation and metastasis [153].

CAF-derived exosomes can carry microRNA, lncRNA, metabolites, and proteins to transmit signals between stromal cells, immune cells, and cancer cells, thus affecting the proliferation and invasion of cancer [154]. Qin et al. [155] found that exocrine-derived Gremlin-1 can promote cancer progression through its regulation of the classical Wnt/β-catenin signal pathway and BMP signal pathway in cancer. In a study by Yan et al. [156], it was discovered that CAF-derived exosome miR-18b plays a role in promoting invasion and metastasis of breast cancer by regulating TCEAL7-induced EMT. Liu et al. [157] discovered that exosomes released by CAFs can enhance the stemness and radiation resistance of colorectal cancer cells through the TGF-β signal pathway. Other studies have found that exocrine miR-3656 secreted by CAFs can promote the occurrence and metastasis of esophageal squamous cell carcinoma via the ACAP2/PI3K-AKT signal pathway [158]. Furthermore, CAF-secreted exosomes possess the ability to promote the neovascularization, thus providing nutritional supply for cancers and promoting the sustained proliferation of cancer cells [159]. Surprisingly, exosomes can also show an inhibitory effect on cancers in some cases. Fujii et al. [160] found that CD9-positive exosomes have the ability to inhibit the progression of malignant melanoma, and the 5-year survival rate of CD9-positive exocrine malignant melanoma patients is higher.

ECM remodeling plays an important part in cancer progression, metastasis, and immunosuppression. CAFs can remodel the ECM by changing the original structure of the ECM to enhance the ECM’s resistance to immune cell infiltration. This alteration aims to inhibit the immune response within the cancer area and increase the immune escape ability of cancers [161]. CAFs can also enhance the immune escape ability of cancers by promoting the infiltration of inhibitory T lymphocytes in the TME and inhibiting the function of effector T cells [162]. Francescone et al. [163] found that NetrinG1 affects the occurrence of pancreatic tumors through nutritional support and the immunosuppression of CAFs.

Solute factors secreted by CAFs, such as IL-6, IL-33, TGF-β, CAF-derived cardiotrophin-like cytokine factor 1 (CLCF1), and stromal cell derived factor-1 (SDF-1), can produce cancer-promoting signals and participate in the signal transduction of cancer cells and other cells within the TME [161]. CAFs can produce various angiogenic factors, including VEGFa, PDGFC, FGF2, WNTs, and MMPs, which can promote cancer angiogenesis by recruiting myeloid cells, vascular endothelial cells, and monocytes combined with unique micro-hypoxia environments [161,164].

Mechanical force also plays an important role in the progression of cancer, especially in cancer invasion and metastasis. It forms a physical connection between CAFs and cancer cells by affecting intracellular mechanical transduction pathways. Mechanical transduction signals in the TME can activate CAFs and enhance their ability to secrete ECM components, thereby promoting the invasive behavior of cancer cells, including proliferation, migration, and invasion [165]. Mechanical forces in the TME can also activate CAF-related signaling pathways, such as the Wnt pathway, thereby enhancing the invasion and metastasis ability of cancer cells [166]. In addition, mechanical force facilitates tight physical interactions between CAFs and cancer cells through intracellular mechanical transduction pathways. It can not only support the movement of cancer cells but also assist cancer cells in invading surrounding tissues through exerting pulling force, promoting the invasion and migration of cancer cells [167,168].

### 4.3. Other Fibroblast-Related Diseases

In addition to playing a key part in the occurrence and progression of fibrosis and cancer, Fbs can also participate in the occurrence and development of some inflammatory diseases and autoimmune diseases together with other immune cells by secreting various cytokines and chemokines, especially chronic inflammatory diseases [169]. Fibroblasts have a pro-inflammatory role in the inflammatory tissue environment. Fibroblasts in tissues such as skin, lung, and kidney can participate in the inflammatory response by secreting cytokines, such as IL-6, IL-8, and MCP1. These cytokines can attract and recruit immune cells, such as neutrophils, lymphocytes, and monocytes [170,171].

The role of synovial fibroblasts (SFs) in joint diseases has gradually attracted attention, especially in inflammatory diseases such as rheumatoid arthritis (RA). Studies have found that SFs actively participate in the process of inflammation maintenance and immune cell recruitment by secreting pro-inflammatory factors, such as IL-6 and TNF-α, in the joint microenvironment. This process aggravates joint damage [172,173]. In addition, SFs have strong migration and invasion abilities and can invade articular cartilage and bone, which is considered an important reason for bone erosion and the pathological hyperplasia of the synovial tissue [174]. These studies suggest the important role of SFs in the development of joint inflammation and provide a basis for the development of potential therapeutic targets for joint inflammatory diseases. For example, the IL-6-Yap-Snail signaling axis is considered to be closely related to the invasive phenotype of SFs. By the targeted inhibition of this signaling pathway, it is expected to slow down the disease progression of inflammatory arthritis [174].

Ankylosing spondylitis (AS) is an autoimmune disease characterized by chronic inflammation and heterotopic ossification. Through the recruitment of immune cells and affecting their differentiation and activation, Fbs can affect the normal immune function of tissues and lead to early inflammation. Fbs promote the articular destruction of cartilage and can lead to osteophyte formation and spinal joint ankylosis through various mechanisms, such as recruiting osteoblasts, directly differentiating into osteoblasts or differentiating into myofibroblasts. The ossification process of Fbs is mediated by the Wnt signal pathway and the BMP/TGF-β signal pathway [175]. Vitiligo is an autoimmune skin disease that is characterized by the depigmentation of the skin and mucosa, leading to the formation of well-defined leukoplakia. Xu et al. [176] found that Fbs in the dermis can recruit and activate CD8+ toxic T cells by secreting some chemokines, which is the main pathogenesis of vitiligo.

## 5. Therapeutic Strategies for Fibroblast-Related Diseases

### 5.1. Antifibrotic Therapy

Fibrosis seriously affects human health. Once fibrosis occurs in organs and tissues, the structural and functional changes caused by fibrosis experience difficulty in returning to the normal state through treatment, and fibrosis causes a considerable medical burden throughout the world. The identification of key therapeutic targets related to fibrosis and seeking effective anti-fibrosis therapy for these targets is the focus of the research on the treatment of fibrosis. However, there is a challenge when considering this treatment; specifically, the treatment that is explored needs to have the ability to combat fibrosis without interfering with the normal homeostasis and wound healing function of the body. Currently, there are mainly two types of treatments for fibrosis: one treatment is for the mechanism of ECM deposition, and the other treatment involves anti-inflammatory therapy. Although many drugs have been proven to have a delaying effect on fibrosis, there is no drug that can completely cure fibrosis.

Pirfenidone and nintedanib have shown efficacy in slowing down the decline in pulmonary function caused by IPF and delay the progression of IPF; moreover, they are well-tolerated by patients. These two drugs have been approved by the Food and Drug Administration (FDA) to treat mild-to-severe IPF, and have been recommended to treat and relieve the IPF in clinical practice guidelines in many countries [177,178,179]. Pirfenidone has been shown to suppress Fb proliferation and the synthesis of collagen and other ECM components [180]. Nintedanib, an intracellular tyrosine kinase inhibitor, targets multiple targets involved in pulmonary fibrosis [181]. While several drugs have shown promise in clinical trials, there is currently no FDA-approved treatment for liver fibrosis, with liver transplantation remaining the sole option for liver cirrhosis. In order to avoid the progression of early liver fibrosis, anti-fibrosis therapy is needed, which is mainly accomplished through etiological treatment, lifestyle intervention, and other forms to prevent and delay the progression of liver fibrosis [182]. Aspirin, a common non-steroidal anti-inflammatory drug, reduces inflammatory cell numbers by inhibiting TNF-α and IL-6, and it can also exhibit a certain anti-inflammatory effect. Additionally, it can hinder the activation and proliferation of hepatic stellate cells by targeting the toll-like receptor 4 (TLR4)/nuclear factor kappa B (NF-κB) signal pathway, thereby slowing down the progression of liver fibrosis [183]. Currently, a significant amount of anti-fibrosis drugs are being researched and tested, some of which show good anti-fibrosis effects in experimental animal models; however, the anti-fibrosis effect is not optimal in clinical trials [184,185].

### 5.2. Anticancer Therapy

Cancers cause serious harm to human health, and the treatment of cancers has always been the focus of research. CAFs are frequently demonstrated to facilitate the occurrence, progression, and metastasis of cancers; so, they also represent a key target for cancer treatment and the prevention of metastasis. There are currently three main anticancer immunotherapy methods targeting CAFs and their related molecules [186,187] (Figure 3).

The most common method for targeting CAF markers involves the inhibitors based on CAF markers, among which the studies on FAP-targeted inhibitors are the most common [188]. At present, the drugs used for targeting the depletion of CAF markers in preclinical or clinical studies mainly include FAP-targeted inhibitors, such as the FAP-targeted DNA vaccine, α-SMA-targeted inhibitors, and PDGFR-targeted inhibitors [186,187]. Currently, a microgenetic vaccine based on the FAP.291 epitope has been demonstrated to perform well in mouse models, as well as induce specific immune responses and inhibit cancer progression [189]. However, to date, the FAP-targeted DNA vaccine has only been used in animal experiments and has not been officially introduced into clinical practice.

The activation and function of CAFs in the TME can be influenced by targeting the effector molecules related to the activation and function of CAFs, including chemokines, growth factors, and cytokines. The TGF-β signal is particularly significant in the activation and function of CAFs, making the inhibition of TGF-β a potential way to improve the compromised immune response in the TME. Bintrafusp alfa (BA) is a fusion protein that concurrently inhibits two immunosuppressive pathways, TGF-β pathway and programmed death-ligand 1 (PD-L1) pathway, potentially impeding cancer cell proliferation by blocking TGF-β signaling [190]. Targeting the TGF-β pathway could offer a promising approach to overcoming chemotherapy resistance in CAFs [191].

The inhibition of ECM proteins derived from CAFs and signaling pathways related to fibrosis activation can inhibit ECM remodeling induced by CAFs, improve ECM hardness, and alleviate the inhibitory effect of the TME on recruiting immune effector cells to a certain extent, thereby alleviating immune suppression [192]. The FAK signal pathway plays a crucial role in the involvement of CAFs in ECM hardness and immune suppression. Studies have found that specific FAK inhibitors (VS-4718) can improve immune suppression in the TEM and improve the overall survival of a mouse pancreatic ductal adenocarcinoma model [193]. TNC is a CAF-derived ECM protein that is produced through selective splicing and protein modification. TNC can regulate cancer immune response and angiogenesis [194]. Murdamoothoo et al. [195] found that TNC can retain CD8 tumor infiltrating T lymphocytes (TILs) in the ECM by binding to CXCL12. When the CXCL12 receptor CXCR4 is blocked, TNC can promote CD8 TILs to enter into the TME, thereby inhibiting cancer growth and metastasis.

## 6. Conclusions

Fbs are among the most prevalent cell types in the human body, and they participate in many important biological processes. Fbs play a vital role in body growth and development, as well as in the maintenance of tissue structure stability. Furthermore, they are also involved in several pathological processes, including organ and tissue fibrosis, cancer proliferation, and metastasis. Therefore, a sufficient understanding of the biological features of Fbs and an exploration of the mechanism of Fbs in pathological processes, as well as actively seeking relevant therapeutic targets, play an important role in delaying disease progression and treating diseases.

Cell markers are often used to distinguish and isolate different types of cells in order to provide targets for cell recognition and treatment of some diseases; however, at present, no specific markers for Fbs have been identified. Since the emergence of single-cell sequencing technology, different subtypes of Fbs and their different functions have been gradually recognized. Traditional cell sequencing techniques usually sequence a group of cells as a whole and mask the subtle differences between individual cells. Single-cell sequencing technology can analyze the genomes and transcriptomes of individual cells, thus revealing the diversity and heterogeneity between cells [196]. The identification of specific markers of Fbs and the active seeking of effective treatments for fibroblast-related diseases (such as fibrosis and cancer) are still the focus of future research.

This paper provides a summary of the recent research advancements in the biological characteristics and physiological functions of Fbs in recent years, as well as focusing on the mechanism and treatment strategies of Fbs in related diseases. It is expected to provide new ideas for further exploring the characteristics of Fbs and for finding related therapeutic targets, as well as aiding in improving the prognosis of patients.

## Figures and Tables

**Figure 1 biomolecules-14-01478-f001:**
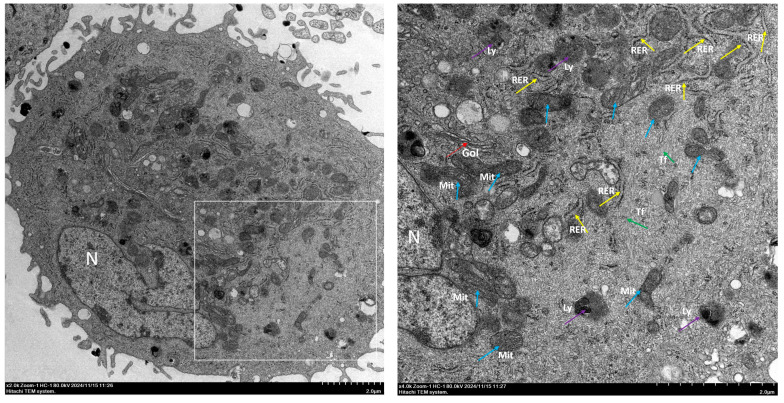
Electron microscopic analysis of human skin fibroblasts. The **left** picture shows the overall imaging of a fibroblast, and the **right** picture is the locally magnified imaging of the left one. Abbreviations: N, Nucleus; RER, Rough endoplasmic reticulum; Gol, Golgi apparatus; Mit, Mitochondrion; Tf, Tonofilament; Ly, Lysosome.

**Figure 2 biomolecules-14-01478-f002:**
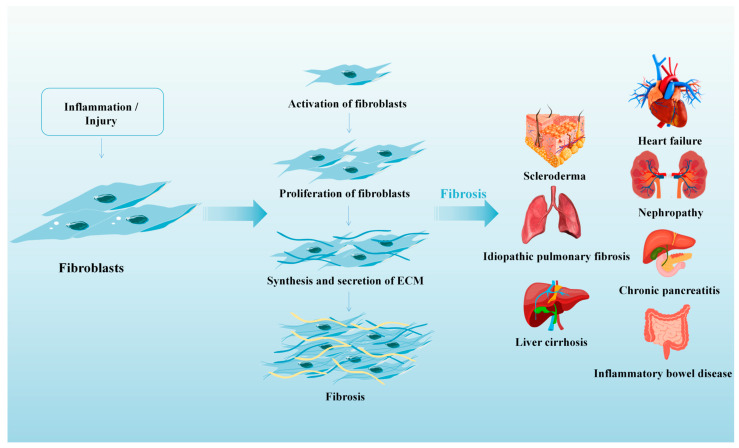
The process of fibrosis and common organs that are related to fibrosis diseases. Idiopathic pulmonary fibrosis (IPF), liver cirrhosis, scleroderma, heart failure, diabetic nephropathy, hypertensive nephropathy, chronic pancreatitis, and inflammatory bowel disease have been found to be associated with fibrosis.

**Figure 3 biomolecules-14-01478-f003:**
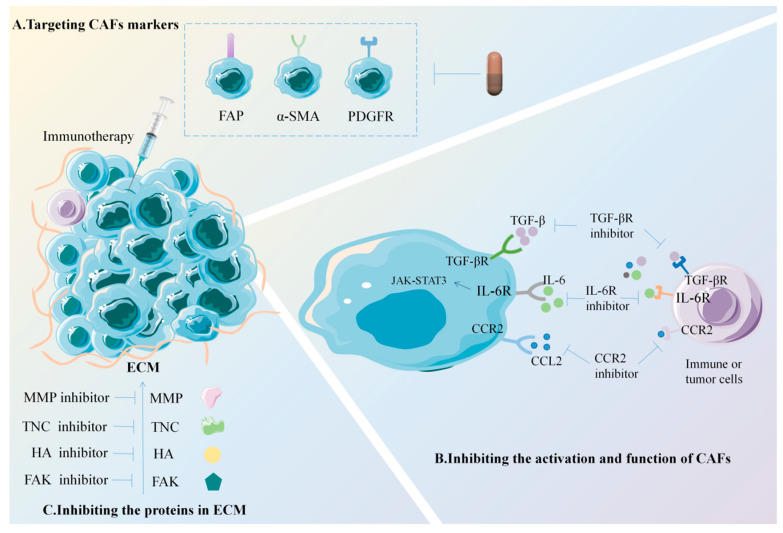
Anticancer immunotherapy methods targeting CAFs and their related molecules. A. The first immunotherapy method is to target CAFs directly by focusing on CAF markers like FAP, PDGFR, and α-SMA to boost the immune response of the body within the TME. B. The second immunotherapy method is to inhibit key signaling pathways or effector molecules related to the activation and function of CAFs, such as IL-6, TGF-β, C–C chemokine ligand 2 (CCL2)-C–C chemokine receptor (CCR2) signal axis, Janus kinase-signal transducer and activator of transcription 3 (JAK-STAT3) signal pathway, and C-X-C chemokine ligand 12 (CXCL12), thus inhibiting the effect of CAFs in the TME. C. The third immunotherapy method is to limit ECM remodeling by inhibiting ECM proteins derived from CAFs, such as MMPs, TNC, hyaluronan (HA), and focal adhesion kinase (FAK) signaling pathways related to fibrosis activation.

**Table 1 biomolecules-14-01478-t001:** Classification of subpopulations of lung Fbs and cardiac Fbs in mice.

The Source of Fbs	Classification of Subpopulations of Fbs
Lung Fbs	Alveolar fibroblasts: Specifically express nephronectin (NPNT) and carboxylesterase 1d (Ces1d) Adventitial fibroblasts: Specifically express protease inhibitor (PI) 16 and decorin (DCN) Peribronchial fibroblasts: Specifically express hedgehog-interacting protein (HHIP), asporin (ASPN), and fibroblast growth factor (FGF) 18
Cardiac Fbs	Fibroblast-Sca1^high^ (F-SH) Fibroblast-Sca1^low^ (F-SL) Fibroblast-transitory (F-Trans) Fibroblast-Wnt expressing (F-WntX) Fibroblast-activated (F-Act)

Lung Fbs in mice can be divided into three subpopulations according to different locations and specific markers, and cardiac Fbs in mice can be divided into five subpopulations according to different functions.

**Table 2 biomolecules-14-01478-t002:** “Universal” FCs and “tissue-specific” FCs in the fibroblast atlas.

Category	Fibroblast Cluster	Tissue Distribution
“universal” FCs	Myh11 SMC Pericyte HSPA6 stress-associated fibroblasts PI16 fibroblast progenitors COL15A1 fibroblast progenitors LRRC15 myofibroblasts SFRP2 myofibroblasts MMP1 myofibroblasts CTNNB1 myofibroblast-like fibroblasts	Widely distributed in multiple organizations
“tissue-specific” FCs	PRG4 fibroblasts SOX6 fibroblasts ADAMDEC1 fibroblasts HHIP SMC IL6 fibroblasts HGF fibroblasts	Synovium Colon and stomach Colon and stomach Colon Breast and lung Kidney and liver

Analysis of fibroblasts by scRNA-seq technology reveals that there are nine “universal” FCs and six “tissue-specific” FCs in the fibroblast atlas. Abbreviations: Myh11, Myosin heavy chain; SMC, Smooth muscle cell; HSPA6, Heat shock protein family A member 6; PI16, Protease inhibitor 16; Col15a1, Alpha1 chain of type XV collagen; LRRC15, Leucine-rich repeat containing 15; SFRP2, Secreted frizzled-related protein 2; MMP1, Matrix metalloproteinase-1; CTNNB1, Catenin beta-1; PRG4, Proteoglycan 4; SOX6, Sex-determining region of Y-related high-mobility-group box 6; ADAMDEC1, A disintegrin and metalloproteinase domain-like decysin 1; HHIP, Human hedgehog-interacting protein; IL6, Interleukin-6; HGF, Hepatocyte growth factor.

**Table 3 biomolecules-14-01478-t003:** Differences in common circulating biomarkers in the fibrosis of different organs.

Organ Fibrosis	Common Circulating Biomarkers	Studied Condition	Test Location	Main Findings	Reference
Pulmonary fibrosis	OPN	ILD patients	Serum	OPN↑	[84]
		IPF patients	Serum	OPN↑	[85]
	MMP-7	IPF patients	Plasma	MMP-7↑	[86]
	ICAM-1	IPF patients	Serum	ICAM-1↑	[87]
	POSTN	IPF patients	Serum	POSTN↑	[88]
Liver fibrosis	HA	Liver fibrosis patients	Serum	HA↑	[90]
		Liver fibrosis and cirrhosis patients	Serum	HA↑	[91]
	LN	Liver fibrosis and cirrhosis patients	Serum	LN↑	[91]
	PIIINP	Liver fibrosis patients	Serum	PIIINP↑	[92]
	Col IV	Liver fibrosis and cirrhosis patients	Serum	Col IV↑	[93]
	CG	Liver fibrosis patients	Serum	CG↑	[94]
Cardiac fibrosis	PICP	HF patients	Serum	PICP↑	[96]
	PINP	HF patients	Serum	PINP↑	[97]
	PIIINP	HF patients	Serum	PIIINP↑	[98]
	CITP	HF patients	Serum	CITP↑	[99]
	MMP-1	HF patients	Serum	MMP-1↑	[100]
	TIMP-1	HF patients	Serum	TIMP-1↑	[101]
	CTGF	HF patients	Plasma	CTGF↑	[102]
	Gal-3	HF patients	Serum	Gal-3↑	[103]
Kidney fibrosis	MCP-1	Children with CKD	Serum	MCP-1↑	[105]
		DKD patients	Plasma	MCP-1 level is positively correlated with DKD progression	[106]
	KIM-1	DKD patients	Plasma	KIM-1 level is positively correlated with DKD progression	[106]
	TNFR-1	DKD patients	Plasma	TNFR-1 level is positively correlated with DKD progression	[106]
	TNFR-2	DKD patients	Plasma	TNFR-2 level is positively correlated with DKD progression	[106]
	MMP-7	IgA nephropathy patients	Serum	MMP-7↑	[107]
	PRO-C3	CKD patients	Serum	PRO-C3 level is positively correlated with CKD progression	[108]
	PRO-C6	Lupus nephritis patients	Serum	PRO-C6 level is positively correlated with renal interstitial fibrosis	[109]

The common circulating biomarkers in pulmonary fibrosis, liver fibrosis, cardiac fibrosis, and kidney fibrosis are different. Abbreviations: ILD, Interstitial lung disease; OPN, Osteopontin; MMP-7, Matrix metallopeptidase-7; ICAM-1, Intercellular adhesion molecule-1; POSTN, Periostin; HA, Hyaluronic acid; LN, Laminin; PIIINP, Procollagen type III N-peptide; Col IV, Type IV collagen; CG, Cholylglycine; HF, Heart failure; PICP, Procollagen type I C-terminal propeptide; PINP, Procollagen type I N-terminal propeptide; CITP, C-terminal telopeptide of collagen type I; MMP-1, Matrix metallopeptidase-1; TIMP-1, Tissue inhibitors of metalloproteinase-1; CTGF, Connective tissue growth factor; Gal-3, Galectin-3 protein; CKD, Chronic kidney diseases; MCP-1, Monocyte chemoattractant protein-1; DKD, Diabetic kidney disease; KIM-1, Kidney injury molecule-1; TNFR-1, Tumor necrosis factor receptor-1; TNFR-2, Tumor necrosis factor receptor-2; Pro-C3, Propeptide of type III collagen; Pro-C6, Propeptide of type VI collagen. “↑” represents an elevated level of circulating biomarkers in the plasma/serum under the research conditions when compared with the control group or normal individuals.

**Table 4 biomolecules-14-01478-t004:** Categories of CAF markers.

Category	CAF Markers
CAF markers with cancer promotion	Fibroblast activation protein (FAP) α-Smooth muscle actin (α-SMA) Periostin (POSTN) Platelet-derived growth factor receptor (PDGFR) Fibroblast-specific protein 1 (FSP-1) Palladin Twist Glutamine-fructose-6-phosphate transaminase 2 (GFPT2) Vimentin Tenascin C (TNC) CD90 CD10 GPR77 Galectin 1 (Gal1) Adipocyte enhancer-binding protein 1 (AEBP1) Osteopontin (OPN)
CAF markers with cancer-restraining functions	Meflin, CD146
CAF markers with bidirectional functions	Caveolin-1 (Cav-1), podoplanin (PDPN), CD200

Cell markers of CAFs can be divided into three categories, including CAF markers with cancer promotion effects, CAF markers with cancer-restraining functions, and CAF markers with bidirectional functions.

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
