# Peer review of "Research Progress of Fibroblasts in Human Diseases"

_biomolecules, 2024, doi:10.3390/biom14111478_

Round 1
Reviewer 1 Report
Comments and Suggestions for Authors
This manuscript provides an interesting survey about the biology of fibroblasts and their role in health and disease. Overall, it is a well written article with some flaws that are summarized below and should be considered by the authors.
Regarding description of the morphology of fibroblast in section 2.1. readers would benefit from some micrographs showing histological or immunohistochemical staining and TEM images of these cells with the features described by the authors. I think such images are mandatory for an article like that because biology is also related to morphology of cells.
In section 2.2. a lot is written about effect of growth factor PDGF but nothing on fibroblast growth factor or transforming growth factors. I think here the authors must add some more details about the effect of these growth factors on fibroblasts like growth induction by FGF-2 ( I think many articles study this phenomenon and the formation of ternary complexes between syndecans, FGF receptors and FGF e.g. ) and others and transition from fibroblasts to myofibroblasts by TGF. Corresponding references should be added and a scheme of other kind of drawing would be desirable.
In section 3, the authors describe matrix synthesis and mention hyaluronan (HA), glycosaminoglycans (GAG), etc. HA represents a GAG and is the only that comes without a protein core. All others are connected to a protein core and named proteoglycans. The remodelling of ECM is occurring by different types of MMPs but not by LOX which is a cross-linker of elastin precursors The authors should also consider that elastin production in fibroblasts is only during perinatal phase and not seen in adult type of cells where it is very limited. (see and cite for example Schmelzer & Duca; The FEBS Journal 289 (2022) 3704–3730).
In 3.2. lines 138-140 the transition of Fb into Myofibroblast during wound healing is mentioned induced by certain cytokines which are presented by TGF beta which should be mentioned here.
In section 4.1.1. the authors describe the process of skin fibrosis in a very general manner in the first paragraph without any reference. In the later part of this section, they provide a general description of the process of fibrosis for different organs by the interplay with macrophages and not specifically for skin. I think the final part should be rather moved to rather to the paragraph of section 4.1. where a more general description of fibrosis is done.
Section 5.3. I have some reservation against the important role of Fb in tissue engineering. For sure in regeneration of dermis they represent a must but in many other applications (e.g. bone, cartilage and other) formation of fibrous/fibrotic tissue is not desired and must be strictly avoided. If you combine Fb with other tissue cells, they would grow typically much faster than the desired cell type and as a result no bone or other tissue is formed. I also would not count here on transdifferentiation (e.g. by use of Fb for making iPSC) because this is a different story. In general, I do not think that this section contributes much to the value of this review unless more detailed information is provided. It should be either removed or more detailed information with pluses and minuses of presence of Fb in tissue engineering applications should be added.
Author Response
First and foremost, we sincerely appreciate the time and effort you have dedicated to reviewing our manuscript and providing such constructive comments. Your professional insights and detailed guidance play a crucial role in improving the quality of the manuscript, and we are deeply grateful for this. We have carefully addressed each of your comments and made corresponding revisions. The main revisions in the new manuscript are as follows:
Comments 1: [Regarding description of the morphology of fibroblast in section 2.1. readers would benefit from some micrographs showing histological or immunohistochemical staining and TEM images of these cells with the features described by the authors. I think such images are mandatory for an article like that because biology is also related to morphology of cells.]
Response 1: Thank you for pointing this out. We agree with this comment. Micrographs of fibroblasts can vividly show the morphological characteristics of fibroblasts and help readers better understand the article. Therefore, We have consulted the relevant database, found the micrograph of fibroblasts, and added it to the manuscript. This figure is located on the page 2, line 75, figure 1 of the revised manuscript.
Comments 2: [In section 2.2. a lot is written about effect of growth factor PDGF but nothing on fibroblast growth factor or transforming growth factors. I think here the authors must add some more details about the effect of these growth factors on fibroblasts like growth induction by FGF-2 ( I think many articles study this phenomenon and the formation of ternary complexes between syndecans, FGF receptors and FGF e.g. ) and others and transition from fibroblasts to myofibroblasts by TGF. Corresponding references should be added and a scheme of other kind of drawing would be desirable.]
Response 2: Thank you for your suggestion. Writing only about PDGF is indeed too one-sided. The effects of fibroblast growth factors or transforming growth factors on fibroblasts should be added. Fibroblast growth factors play an important role in the proliferation, migration and differentiation of fibroblasts. Therefore, We have consulted the latest literature on fibroblast growth factors and transforming growth factors and organized the content. this change can be found – page 5, lines 152-172.
Comments 3: [In section 3, the authors describe matrix synthesis and mention hyaluronan (HA), glycosaminoglycans (GAG), etc. HA represents a GAG and is the only that comes without a protein core. All others are connected to a protein core and named proteoglycans. The remodelling of ECM is occurring by different types of MMPs but not by LOX which is a cross-linker of elastin precursors The authors should also consider that elastin production in fibroblasts is only during perinatal phase and not seen in adult type of cells where it is very limited. (see and cite for example Schmelzer & Duca; The FEBS Journal 289 (2022) 3704–3730).]
Response 3: Thank you very much for correcting the content. We are very sorry for this mistake. Indeed, we did not understand this point deeply enough. After consulting relevant literature, we have gained a new understanding of glycosaminoglycans and the extracellular matrix remodeling process. In addition, elastin only occurs during the perinatal period and is very limited in adult cells. After referring to the literature recommended by the reviewers, we also have a new understanding of this content and describe this content in the manuscript. this change can be found – page 5, lines 181-183, 190.
Comments 4: [In 3.2. lines 138-140 the transition of Fb into Myofibroblast during wound healing is mentioned induced by certain cytokines which are presented by TGF beta which should be mentioned here.]
Response 4: Thank you for pointing this out. We agree with this comment. TGF plays an important role in the process of fibroblast transformation into myofibroblasts. TGF participates in this process by presenting cytokines. I have added this content to the revised manuscript. This change can be found – page 6, line 215.
Comments 5: [In section 4.1.1. the authors describe the process of skin fibrosis in a very general manner in the first paragraph without any reference. In the later part of this section, they provide a general description of the process of fibrosis for different organs by the interplay with macrophages and not specifically for skin. I think the final part should be rather moved to rather to the paragraph of section 4.1. where a more general description of fibrosis is done.]
Response 5: Thank you very much for your suggestion. In section 4.1.1, the description of skin fibrosis is indeed somewhat general and some specific content should be added. The content of the interaction between fibroblasts and macrophages is more suitable for describing fibrosis as a whole rather than being limited to skin fibrosis. It is indeed more appropriate to place it in section 4.1. Thank you very much for your suggestion. I have adjusted the content of this paragraph and the positions of related contents. This change can be found – page 8, lines 320-326, line 334 and page 9, lines 340-343.
Comments 6: [Section 5.3. I have some reservation against the important role of Fb in tissue engineering. For sure in regeneration of dermis they represent a must but in many other applications (e.g. bone, cartilage and other) formation of fibrous/fibrotic tissue is not desired and must be strictly avoided. If you combine Fb with other tissue cells, they would grow typically much faster than the desired cell type and as a result no bone or other tissue is formed. I also would not count here on transdifferentiation (e.g. by use of Fb for making iPSC) because this is a different story. In general, I do not think that this section contributes much to the value of this review unless more detailed information is provided. It should be either removed or more detailed information with pluses and minuses of presence of Fb in tissue engineering applications should be added.]
Response 6: Thank you for your suggestion. The application of fibroblasts in tissue engineering is indeed a matter of controversy. Currently, there are relatively few research studies on the application of fibroblasts in the tissue - engineering process. When it comes to the application of fibroblasts in many tissues, there exist certain deficiencies and drawbacks. Additionally, this section makes little contribution to the value of this article. Thank you very much for your valuable suggestion. I have already deleted the content of this section. This change can be found – page 16, lines 656-672.
Reviewer 2 Report
Comments and Suggestions for Authors
In this review article the authors attempt to handle a really vast object, i.e the contribution of fibroblasts in human diseases and the recent advances thereof.
After a brief introduction to fibroblast identification history they refer to some differentiation and proliferation qualities and contribution in homeostasis of this cell type, before entering the main theme: wound healing, regeneration and of course several types of organ fibrosis (skin, lung, liver). In addition, they examine the fibroblast (CAF) contribution in cancer development and progression. Some anti-fibrotic (and anticancer) strategy examples are given at the end along with a brief introduction to engineered tissue approaches to target disease. This is a strong attempt and it is understood and accepted that such a review cannot include all aspects of fibroblast biology and relevant research progress. However, there are some important gaps concerning areas that are under- or not at all represented that need to be addressed in this context.
First of all there is very limited reference to the tremendous progress made in several organs as regards fibroblast heterogeneity, based mainly in single cell RNA Seq. When combined with spatial transcriptomics or other multiomics approaches it yielded several important data sets showing among others differential actions and functions of fibroblast subtypes. There are various examples (e.g. among others PMIDs 32317643, 35948637, 37291214) that should be mentioned and ideally organized in a table. Cross tissue examples identifying similarities or differences among organs are also underway and missing too (e.g. PMIDs 35649411, 33981032, 35649411). Notably, among the recent progress concerning fibroblasts the important research that has been conducted in cardiac and kidney fibrosis is lacking and should be at least briefly described. In addition, the proinflammatory identities of fibroblasts (see for instance PMID 33911232) are underrepresented including recent progress in synovial fibroblasts.
As the article starts with the basics of fibroblast identity a brief mention of the features allowing discrimination between fibroblasts and mesenchymal stem (stromal) cells (SMCs) and other fibroblast-like cells, e.g. telocytes, would be expected in the introductory part.
Figure 1 is too general and much less informative than needed for such a review. It would be better to replace it with a figure depicting the major pathways affected that are important for organ fibrosis and mentioned in the text. In addition, a table showing fibroblast markers in human disease, in general not only cancer, would add a lot.
Author Response
First and foremost, we sincerely appreciate the time and effort you have dedicated to reviewing our manuscript and providing such constructive comments. Your professional insights and detailed guidance play a crucial role in improving the quality of the manuscript, and we are deeply grateful for this. We have carefully addressed each of your comments and made corresponding revisions. The main revisions in the new manuscript are as follows:
Comments 1: [First of all there is very limited reference to the tremendous progress made in several organs as regards fibroblast heterogeneity, based mainly in single cell RNA Seq. When combined with spatial transcriptomics or other multiomics approaches it yielded several important data sets showing among others differential actions and functions of fibroblast subtypes. There are various examples (e.g. among others PMIDs 32317643, 35948637, 37291214) that should be mentioned and ideally organized in a table. Cross tissue examples identifying similarities or differences among organs are also underway and missing too (e.g. PMIDs 35649411, 33981032, 35649411).]
Response 1: Thank you for pointing this out. We agree with this comment. Single-cell RNA sequencing technology plays a significant role in revealing the diversity and heterogeneity of cells and has also led to tremendous progress in the research of fibroblasts. We should describe in the article the research progress of single-cell RNA sequencing technology in fibroblasts, including the subtype heterogeneity and organ heterogeneity of fibroblasts. I have read the references provided by experts and consulted the latest literature on single-cell RNA sequencing of fibroblasts, made a summary and added the content to the manuscript. This change can be found – Section 2.2. and table 1 and table 2.
Comments 2: [Notably, among the recent progress concerning fibroblasts the important research that has been conducted in cardiac and kidney fibrosis is lacking and should be at least briefly described.]
Response 2: Thank you for your suggestion. We agree with this comment. Cardiac fibrosis and kidney fibrosis have a great impact on human health, severely affecting people's quality of life and having a poor prognosis. Cardiac fibrosis and kidney fibrosis should be described in this article, with the expectation of providing references for readers to fully understand fibroblasts and fibrosis. I have already read the latest literature on cardiac and kidney fibrosis and supplemented this part of the content in the manuscript. This change can be found – Section 4.1.4 and Section 4.1.5.
Comments 3: [In addition, the proinflammatory identities of fibroblasts (see for instance PMID 33911232) are underrepresented including recent progress in synovial fibroblasts.]
Response 3: Thank you for your insightful suggestion. Fibroblasts have a pro - inflammatory effect in the inflammatory tissue environment. Moreover, fibroblasts can participate in the occurrence and development of some inflammatory and autoimmune diseases together with other immune cells by secreting various cytokines and chemokines. Currently, the role of synovial fibroblasts in joint diseases has also received attention. We have read the literature recommended by the experts. By combining it with the latest literature regarding the proinflammatory identities of fibroblasts, we have supplemented the content. This change can be found – page 13, lines 542-561.
Comments 4: [As the article starts with the basics of fibroblast identity a brief mention of the features allowing discrimination between fibroblasts and mesenchymal stem (stromal) cells (SMCs) and other fibroblast-like cells, e.g. telocytes, would be expected in the introductory part.]
Response 4: Thank you very much for your suggestion. Fibroblasts have similar characteristics in some aspects with mesenchymal stem cells and fibroblast - like cells. Therefore, understanding the differences between fibroblasts and other cells is of great significance for a more comprehensive understanding of fibroblasts. After reading relevant information, we have summarized and sorted out the differences and connections between fibroblasts, mesenchymal stem cells and fibroblast - like cells. This change can be found – page 1, lines 25-37.
Comments 5: [Figure 1 is too general and much less informative than needed for such a review. It would be better to replace it with a figure depicting the major pathways affected that are important for organ fibrosis and mentioned in the text.]
Response 5: Thank you for your suggestion. We agree with this comment. Figure 1 only presents several important fibrotic organs and fails to demonstrate the process of organ fibrosis. A display of the fibrosis process should be provided to facilitate readers' better understanding of fibrosis. I have redesigned and drawn a picture, in which the process of fibrosis occurrence is presented. This change can be found – page 8, line 335, figure 2.
Comments 6: [In addition, a table showing fibroblast markers in human disease, in general not only cancer, would add a lot.]
Response 6: Thank you for your insightful suggestion. Not only in cancer, but also in fibrotic diseases of different organs, there are different fibroblast biomarkers, and these biomarkers may become important targets for fibrotic diseases. They are of great significance for the diagnosis and treatment of fibrotic diseases. I have read the relevant literature and supplemented this content. This change can be found – page 9, line 346, table 3.
Reviewer 3 Report
Comments and Suggestions for Authors
see attached

Author Response
First and foremost, we sincerely appreciate the time and effort you have dedicated to reviewing our manuscript and providing such constructive comments. Your professional insights and detailed guidance play a crucial role in improving the quality of the manuscript, and we are deeply grateful for this. We have carefully addressed each of your comments and made corresponding revisions. The main revisions in the new manuscript are as follows:
Comments: [While it indirectly refers to the role of fibroblasts related to several mechanotranduction pathways, it overlooks the role of mechanical forces in affecting fibroblast behavior in health and disease. The role of fibroblasts in mechanotransduction and the effects of mechanical forces on fibroblast behavior and fibroblast ECM interactions cannot be overlooked since much work has been done on fibroblast contraction of collagen lattices, the role of mechanical forces in skin, tendon and ligament behavior, cell stiffness and tissue homeostasis. I recommend the authors revise their manuscript to cover the influence of mechanical forces on fibroblast behavior and how this relates to mechanotransduction in health and disease. Without this connection the review provides a limited perspective on fibroblast behavior and its influence on tissue homeostasis and disease progression.]
Response: Thank you for pointing this out. We agree with this comment. Mechanical forces play an important role in maintaining tissue homeostasis, wound healing and pathological responses. Moreover, mechanical forces have a significant impact on fibroblasts as they are important regulators of fibroblast function and behavior. When I review fibroblasts, we should introduce in - depth the influence of mechanical forces on fibroblasts in health and disease, which will help readers better and more comprehensively understand the characteristics and functions of fibroblasts. I have already read the latest literature on the relationship between mechanical forces and fibroblasts, and made some summaries. I have added the summary content to the manuscript. This change can be found – page 7, lines 260-277 and page 8, lines 327-333 and page 13, lines 528-539.
Round 2
Reviewer 1 Report
Comments and Suggestions for Authors
The authors improved her review paper substantially and made it highly interesting for the readers. I have no further comments and suggest acceptance of this very nice manuscript.
Author Response
Comments: [The authors improved her review paper substantially and made it highly interesting for the readers. I have no further comments and suggest acceptance of this very nice manuscript.]
Response: Thank you so much for your kind and positive comments. We are extremely grateful for the revision suggestions you provided during the review process previously, which can improve the quality of the article. We also highly appreciate your recognition of the improvements made in our manuscript. Your suggestion to the journal for its acceptance means a great deal to us. We appreciate your time and effort in reviewing our work.
Reviewer 2 Report
Comments and Suggestions for Authors
In the revised version the authors addressed all issues raised and considerably improved their manuscript by providing new text that summarizes findings and concepts regarding renal and cardiac fibrosis, biomarkers of fibrosis, fibroblast heterogeneity, as well as fibroblast functions in inflammatory conditions. In addition, sections considering the contribution of mechanosensing were added. They further support their rationale by two new tables and two new figures. Some of the additions in this improved version need however a couple of minor modifications and clarifications.
Table 1 presents lung and cardiac fibroblast subdivisions according to two recent and comprehensive scRNASeq-based studies (Refs 15 and 16). Since numerous attempts were conducted analyzing organ fibroblasts at the single cell level and the studies mentioned are largely based on fate mapping settings in the mouse while other classifications may exist in the human (for instance PMIDs 32971526, 47438528 and 35948637 for human heart) it should be emphasized that are these are examples primarily addressing the mouse organ identities.
The scope of including figure 1 is not clear. In any case the legend should be more descriptive clearly indicating the different organelles or cytoplasmic structures shown.
In Table 3 it is stated that common biomarkers in fibrosis are listed which also differ between particular organs. Some modifications like an altered title and legend should be included to better describe the authors’ view. For instance, it is not true that peptides from cleaved collagen are biomarkers only in liver and kidney fibrosis (for instance PMID 33014530). In addition it is not clear what is meant with the term biomarkers. Does it refer to circulating biomarkers, or tissue indicators of fibrosis, or both? Also some of the references used in that table indicate a mediator rather than a biomarker (e.g. Ref 86, 87). In that case other molecules may be included as well. The authors could facilitate the reader by clarifying these points and furthermore present this table as one citing some examples rather than being a comprehensive list.
Author Response
Comments 1: [Table 1 presents lung and cardiac fibroblast subdivisions according to two recent and comprehensive scRNASeq-based studies (Refs 15 and 16). Since numerous attempts were conducted analyzing organ fibroblasts at the single cell level and the studies mentioned are largely based on fate mapping settings in the mouse while other classifications may exist in the human (for instance PMIDs 32971526, 47438528 and 35948637 for human heart) it should be emphasized that are these are examples primarily addressing the mouse organ identities.]
Response 1: Thank you for pointing this out. We agree with this comment. There are differences in the classification of fibroblasts in mouse organs and that in human organs. We should clearly point out in the manuscript that the research mentioned is mainly based on studies of mice, so as to enable readers to understand the article more accurately. We have already added an explanation in the corresponding positions that the research mentioned is based on studies of mice. Besides, we have also incorporated a description that the classification of fibroblasts in human organs differs from that in mice. This change can be found – page 3, Section 2.2. and table 1.
Comments 2: [The scope of including figure 1 is not clear. In any case the legend should be more descriptive clearly indicating the different organelles or cytoplasmic structures shown.]
Response 2: Thank you for your suggestion. We agree with this suggestion. Providing clear and descriptive legends contributes significantly to readers' comprehension of the article content. We have created the electron - microscope images of fibroblasts and performed local magnification processing on these images. Moreover, the significant organelle structures have been annotated within the magnified images. We appreciate the valuable suggestion. This change can be found – page 2 Figure1.
Comments 3: [In Table 3 it is stated that common biomarkers in fibrosis are listed which also differ between particular organs. Some modifications like an altered title and legend should be included to better describe the authors’ view. For instance, it is not true that peptides from cleaved collagen are biomarkers only in liver and kidney fibrosis (for instance PMID 33014530). In addition it is not clear what is meant with the term biomarkers. Does it refer to circulating biomarkers, or tissue indicators of fibrosis, or both? Also some of the references used in that table indicate a mediator rather than a biomarker (e.g. Ref 86, 87). In that case other molecules may be included as well. The authors could facilitate the reader by clarifying these points and furthermore present this table as one citing some examples rather than being a comprehensive list.]
Response 3: Thank you for your insightful suggestion. We agree with your suggestion. The biomarker this article intends to describe refers to the circulating biomarker. We are truly sorry for not clearly indicating it in the manuscript. We have thoroughly read the literature recommended by the reviewers and consulted other relevant literature, gaining a further understanding of the circulating biomarkers of organ fibrosis and making supplements to the content in the manuscript. In addition, in accordance with your suggestions, the tables have been revised and redrawn, thus enabling readers to view and understand the content better and more clearly. We are deeply indebted for your patience and expert guidance, as well as for your invaluable assistance in enhancing the quality of the manuscript. This change can be found – page 9, lines 348-362 and Table 3.
Reviewer 3 Report
Comments and Suggestions for Authors
The manuscript is much improved and is better focused. Mechanotransduction is also covered as was requested in reviews.
Comments on the Quality of English LanguageThe manuscript is much improved and is better focused. Mechanotransduction is also covered as was requested in reviews.
Author Response
Comments 1: [The manuscript is much improved and is better focused. Mechanotransduction is also covered as was requested in reviews.]
Response 1: Thank you very much for your kind recognition of the significant improvement in our manuscript during the revision stage. Your professional review and valuable suggestions have been of great help to us. We truly appreciate your time and effort.
Comments 2: [Minor editing of English language required.]
Response 2: Thank you for pointing this out. We agree with this comment. I have already checked the English language expressions in the manuscript content and made revisions and edits. This change can be found – page 7, lines 267-282; page 9, line342-343; page 14, lines 539-548.